# End-to-End Patch-Based Semantic Segmentation for Paramagnetic Rim Lesion Detection

**Zineb El yamani**[1]                       ZINEB.EL.YAMANI@USHERBROOKE.CA
**Antoine Théberge**[1]
**Manon Edde**[1]
[1] *Sherbrooke Connectivity Imaging Laboratory, University of Sherbrooke, QC, Canada*
**Olivier Grimard**[2]
**Emmanuelle Lapointe**[2]
[2] *Department of Medicine, Division of Neurology, University of Sherbrooke, QC, Canada*
**Stefano Magon**[3]
**Muhamed Barakovic**[3]
[3] *Roche Pharma Research and Early Development, Neuroscience and Rare Diseases, Roche Innovation Center Basel Switzerland, F. Hoffmann-La Roche Ltd., Basel, Switzerland*
**Maxime Descoteaux**[1]
**François Rheault**[1]

**Editors:** Accepted for publication at MIDL 2025

## Abstract

Paramagnetic Rim Lesions (PRLs) are an emerging biomarker of chronic active inflammation in Multiple Sclerosis (MS) but their visual identification on susceptibility-sensitive MRI remains challenging and time-intensive. Due to the scarcity of PRLs, existing automated methods rely on patch-based classification, where a lesion-centered 3D patch is classified as PRL or non-PRL. However, MS lesions often occur in clusters, so a single patch may contain multiple types of lesions. Moreover, this approach requires prior extraction of lesion-centered patches which complicates the reconstruction of whole-brain predictions. To overcome this, we propose an end-to-end, whole-brain pipeline that generates patches on the fly and directly delineates PRLs within them, eliminating the need for lesion-centered extraction and enabling more precise and user-friendly automated PRL detection.

**Keywords:** Multiple Sclerosis, Chronic Active Lesions, Paramagnetic Rim Lesions, Segmentation, Magnetic Resonance Imaging, Quantitative Susceptibility Mapping.

## 1. Introduction

Multiple Sclerosis (MS) is a chronic inflammatory disease of the central nervous system marked by demyelinating lesions visible on T2-FLAIR MRI. Paramagnetic rim lesions (PRL), identified by iron-enriched rims on susceptibility-sensitive MRI (Bagnato et al., 2024), are emerging as key biomarkers of the disease and are expected to become part of standard clinical assessments according to 2024 revision of the McDonald criteria (Montalban, 2024). However, manual PRL identification is labor-intensive and impractical for clinical use. Deep learning models have been proposed for PRL detection using lesion-centered 3D patch classification (Barquero et al., 2020; Zhang et al., 2022), but struggle with clustered

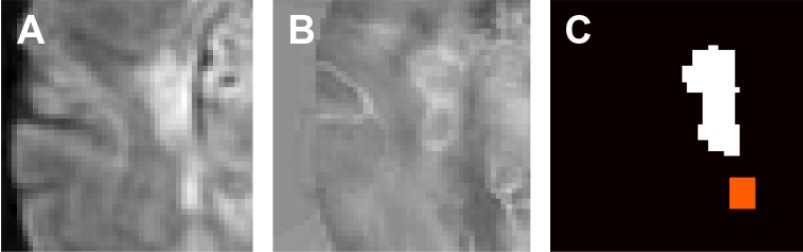

Figure 1: Cluster of multilabel lesions. (A) T2-FLAIR image showing lesions as hyperintense spots; (B) Corresponding QSM image highlighting the hyperintense rims of two confluent PRLs; (C) Expert annotation map. White indicates PRL, red indicates non-PRL.

lesions (c.f. Figure 1) and requires extensive pre- and post-processing. Whole-brain segmentation offers an interesting alternative (Wynen et al., 2024) but has shown limited performance due to small datasets. To address these issues, we propose a fully automated patch-based semantic segmentation method that directly delineates PRLs and non-PRLs, improving precision and eliminating the need for pre-computing lesion-centered inputs in a low-data setting. The code for our method is available at: https://github.com/zinelya/PRLseg.

## 2. Participants and proposed method

**Data** We used a subset of preprocessed imaging data from a longitudinal in-house database, including 20 patients with relapsing-remitting multiple sclerosis (RRMS) recruited from the neurology clinic at the Centre Hospitalier Universitaire de Sherbrooke, Canada. Each patient underwent five MRI sessions over 5–6 months, with 4-week intervals. All scans were acquired on a 3T scanner and include T2-FLAIR and Quantitative Susceptibility Mapping (QSM) images reconstructed from multi-echo GRE sequences. Detailed acquisition parameters are available in Appendix A. Lesions, without PRL status, were segmented using the NeuroRx [1] machine learning tool and uniquely labeled with Scilpy [2]. FLAIR and lesion masks were rigidly registered to QSM magnitude images. The paramagnetic status of each lesion was assessed by a neurology resident trained in PRL recognition. Of 1,123 segmented lesions, 73 were identified as PRLs. Ratings were done for all lesions in the first session and for new lesions in later sessions, with the assumption that PRLs in session 1 were still paramagnetic in subsequent sessions due to short scanning intervals and the chronic nature of PRLs. A subject-wise stratified split of 12, 4 and 4 subjects for training, validation and testing, was used to prevent data leakage across timepoints and ensure a balanced distribution of PRLs across sets.

**Method** Our method performs patch-wise lesion segmentation from T2-FLAIR and QSM images. We input $32 \times 32 \times 32$ patches of coregistered T2-FLAIR and QSM to a neural network architecture and output two binary masks representing non-PRL and PRL lesions respectively. As lesion masking is part of the MS clinical routine, we also explore adding

---

1. https://neurorx.com/en
2. https://www.github.com/scilus/scilpy

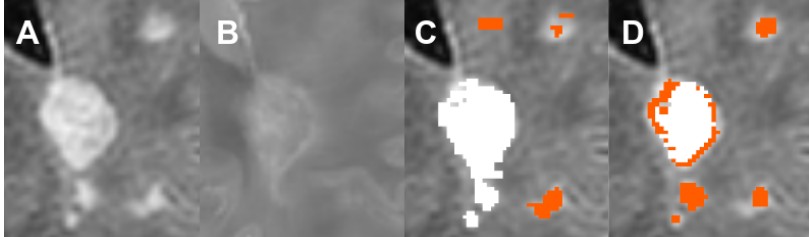

Figure 2: Comparative example from the test set. A) FLAIR image. B) QSM image. C) Expert annotations. D) Aggregated segmentation output from the UNet. White indicates PRL, red indicates non-PRL.

the binary lesion mask as input. During training, per epoch, three patches per subject are generated at random, centered around a voxel in the ground-truth lesion mask. We explore using a 3D UNet (Ronneberger et al., 2015) and a 3D UNETR (Hatamizadeh et al., 2022) to perform our segmentation (c.f. section 3). At inference, patches are extracted using grid sampling and processed independently. Predictions are aggregated to reconstruct the whole-brain output.

## 3. Experiment and results

To evaluate the accuracy of our proposed method, we train and validate our method on our in-house dataset. Networks are trained for 3000 epochs using a learning rate of $1e - 4$. We report the global F1 score, along with Dice scores specific to PRL and non-PRL classes on the testset.

| Model | F1 | Dice PRL | Dice non-PRL |
|---|---|---|---|
| UNet | **0.4976** $\pm$0.0830 | 0.2573 $\pm$0.2457 | **0.2502** $\pm$0.1772 |
| UNETR | 0.3890 $\pm$0.0929 | 0.3260 $\pm$0.2744 | 0.0353 $\pm$0.0416 |
| UNet$_{lesion}$ | 0.4603 $\pm$0.0881 | **0.4223** $\pm$0.2989 | 0.0528 $\pm$0.0500 |
| UNETR$_{lesion}$ | 0.3271 $\pm$0.0854 | 0.3268 $\pm$0.2606 | 0.0112 $\pm$0.0136 |

Table 1: Performance comparison of different 3D models on lesion segmentation.

As we can observe from Table 1, although the final Dice scores are quite low, the UNet architecture clearly outperforms UNETR. Interestingly, including the lesion mask as part of the input does not seem to improve the final segmentation.

## 4. Discussion, conclusion and future works

Our work highlights the challenges of PRL segmentation due to the scarcity of data and unreliable data annotation. As shown in Figure 2, while it includes spurious erroneous voxels, the predicted lesion masks seem to be more in line with the FLAIR hyperintensities than the ground-truth. For instance, the model segments the large lesion into distinct PRL and non-PRL regions. Future work should focus on larger, multi-rater annotated datasets that also explore the use of other susceptibility-sensitive modalities.

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

## Appendix A. Acquisition parameters

|                                      | 3D T2 FLAIR                    | 3D mGRE                           |
| ------------------------------------ | ------------------------------ | --------------------------------- |
| Resolution (mm)                      | $0.74 \times 0.74 \times 0.56$ | $0.79 \times 0.79 \times 0.8$     |
| Repetition time (TR, ms)             | 4800                           | 35.39                             |
| Inversion Recovery Time (TI, ms)     | 1650                           | -                                 |
| Number of echoes                     | 4                              | 4                                 |
| Echo time (TE, ms)                   | 340                            | $TE_1 = 4.96$, $\Delta TE = 9$    |
| Flip angle (°)                       | 40                             | 17                                |
| Scan time                            | 4'33"                          | 4'36"                             |

Table 2: MRI acquisition protocol on a 3T Philips Ingenia scanner.

