# OpenReview forum: "End-to-End Patch-Based Semantic Segmentation for Paramagnetic Rim Lesion Detection"
_MIDL.io/2025/Short_Papers — MIDL 2025 - Short Papers_

### Official Review · Reviewer_YosB · 2025-04-29

**Rating:** 4
**Confidence:** 5

**Summary:**

This paper proposes a patch-based paramagnetic rim lesion (PRL) and non-PRL segmentation method using T2-FLAIR and QSM images as input. A binary lesion mask is also incorporated as an additional input. Two network architectures, UNet and UNETR, are evaluated for performance.

**Strengths:**

•	PRL segmentation and detection is a challenging and important task in automated MS lesion analysis. This work explores the use of patch-based training and inference to address this problem.

**Weaknesses:**

•	A major weakness is the lack of comparison to existing benchmark PRL detection methods, such as QSMRim-Net (Zhang, Hang, et al., NeuroImage: Clinical, 2022).
•	Additionally, a comparison between phase-based PRL detection methods and QSM-based methods would further strengthen the evaluation.

---

### Decision · Program_Chairs · 2025-05-01

Accept